

# A survey on common criteria (CC) evaluating schemes for security assessment of IT products

Maheen Fatima[1], Haider Abbas[1], Tahreem Yaqoob[1], Narmeen Shafqat[1], Zarmeen Ahmad[1], Raja Zeeshan[1], Zia Muhammad[1], Tauseef Rana[2] and Shynar Mussiraliyeva[3]

[1] Department of Information Security, National University of Sciences and Technology, Islamabad, Pakistan
[2] Department of Software Engineering, National University of Sciences and Technology, Islamabad, Pakistan
[3] Al-Farabi Kazakh National University, Almaty, Kazakhstan

Corresponding author
Haider Abbas, haider@mcs.edu.pk

## ABSTRACT

Over the last few years, private and public organizations have suffered an increasing number of cyber-attacks owing to excessive exploitation of technological vulnerabilities. The major objective of these attacks is to gain illegal profits by extorting organizations which adversely impact their normal operations and reputation. To mitigate the proliferation of attacks, it is significant for manufacturers to evaluate their IT products through a set of security-related functional and assurance requirements. Common Criteria (CC) is a well-recognized international standard, focusing on ensuring security functionalities of an IT product along with the special emphasis on IS design and life-cycle. Apart from this, it provides a list of assurance classes, families, component, and elements based on which security EALs can be assigned to IT products. In this survey, we have provided a quick overview of the CC followed by the analysis of country-specific implementation of CC schemes to develop an understanding of critical factors. These factors play a significant role by providing assistance in IT products evaluation in accordance with CC. To serve this purpose, a comprehensive comparative analysis of four schemes belonging to countries including US, UK, Netherlands, and Singapore has been conducted. This comparison has aided to propose best practices for realizing an efficient and new CC scheme for the countries which have not designed it yet and for improving the existing CC schemes. Finally, we conclude the paper by providing some future directions regarding automation of the CC evaluation process.

## INTRODUCTION

The rapid advancements in the field of IT bring opportunities as well as new challenges in the form of associated cyber threats, technological vulnerabilities, and security risks. These misconfigurations and software flaws make IT products open to attacks and exploitation (*Rastogi, 2019*; *Costa, 2019*). During the last two decades, security threats have been growing exponentially because of the evolution of IT products and lack of secure

development practices by developer; thereby introducing major concerns for consumers, organizations, regulators, and government authorities. Attacks such as malware infections, DOS, data tampering, APTs, side channel attack exploit etc. exploit the vulnerabilities of IT products to gain financial benefits. According to Insider Threat Report 2018 (*Schulze, 2018*), 50% of the database systems and 46% of the file systems became major target of cyber attacks in 2018.

Today government organizations and industry are highlighting information assurance and cybersecurity as one of their top priorities. Cyberattacks are performed by both nation-sponsored groups and individuals to gain trade secrets, conduct espionage, and remotely access IT products (*Faizi, 2019*). Moreover, new issues have been increasing regarding supply chain integrity with counterfeit and tampering incidents, which eventually degrade the confidence of users. The organizations experiencing these cyber-attacks face loss of confidential information due to which they endure penalties of millions of dollars and huge business losses (*Costa, 2019*). Over the decades, researchers and industries have been working together to have a globally accepted evaluation standard to evaluate IT products thoroughly and address security vulnerabilities at early stages (*Insua, 2019*). However, in such an innovative, mobile, and collaborative age, the process of achieving secure IT infrastructure and products has become a great challenge. Characterizing an IT product to be secure requires robust criteria against which a system can be evaluated (*El-Hadary, 2014*; *Bialas, 2017*). At a minimum, security evaluation standards provide a set of security requirements, which assist in developing a baseline for evaluating an IT product relative to its specifications (*Katt, 2019*).

Over the years, several standards have been adopted worldwide in order to evaluate IT products. However, Common Criteria (CC) has emerged as a unified international standard targeting both military and commercial needs. Evaluation of IT products in compliance with CC ensures the reduction of security vulnerabilities, as generally it has been observed that non-certified products have a relatively larger number of vulnerabilities as compared to their certified counterparts. Moreover, the significant advantages of this international harmonized evaluation standard include but not limited to; enhancement of accessibility of security-strengthened and evaluated IT products, increasing the confidence of consumers, maximizing the capability and cost-effectiveness of the certification and evaluation procedures, permitting vendors to emphasize on the resources regarding standard requirements for enhancing security in IT products, and increasing the number of secure and certified products. However, to conduct evaluations through CC, a state must be its authorizing member. For a state to become an authorizing member, a procedure to develop and certify a scheme, CB and evaluation facility must be followed, which is troublesome. Majority of the third world countries find it extremely challenging to implement the whole procedure of developing CC scheme, CB, and evaluation facility. In this regard, this survey studies and analyzes the approaches adopted by countries with high security ranking for implementing CC. Moreover, the best practices and lessons learnt from four CC-certificate authorizing members (US, UK, Netherlands, and Singapore) are studied and analyzed to develop guidelines that can be adopted by developing countries to conduct CC evaluations.

## Paper organization

The rest of the research is organized in seven different sections which discuss all major contributions of paper. 'Related Work' presents a summary of relevant literature. 'Research Methodology' presents the research methodology employed for carrying out the survey. 'Survey of Country Level Implementation of CC Schemes' discusses in detail the country level implementation of CC schemes of four developed as well as CC-certificate authorizing states; US, UK, Singapore and Netherlands. 'Comparative Analysis' presents a comparative analysis of the studied four CC schemes. 'Best Practices to be Followed while Establishing a New Country Scheme' proposes best practices to be followed while establishing a new country scheme. Lastly 'Conclusion' concludes the paper and 'Open Challenges' presents open research areas.

## RELATED WORK

Some of the most relevant work related to testing, evaluation methodology, and certification procedure of CC is discussed below.

*Hong & Kim (2013)* compared the security status of the IT security solution before and after the CC evaluation and analyzed the results. A questionnaire was designed for the domestic solutions vendors. By using statistical analysis, it had been illustrated that CC has positively affected the security of IT solutions quantitatively. This research was aimed to enhance the security of domestic security solutions.

Significance of CC in the perspective of secure software development is also highlighted in literature. In this regards, Mehmat Kara has surveyed existing secure software development standards and models (*Kara, 2012*). After a detailed review, it was concluded that CC in addition to evaluation of IT products can be used as a guidance for secure software development life-cycle for software developers as it provides a holistic set of necessary requirements. Moreover, it was proposed that addition of security functions like procedure compliance, policy, and law to the future version of CC will result in more secure products. *Mellado, Fernández-Medina & Piattini (2007)* have presented a CC-centered and reuse-based process for dealing with security requirements in a structured and intuitive manner during the initial phases of software development. Authors have aimed to unify the concepts of requirements engineering and security engineering by providing a security resources repository and incorporating CC into the software life-cycle.

Several contributions to improve the assurance scheme and evaluation and certification process of CC were found in the existing literature which are discussed. *Bialas (2018)* discusses the CC assurance methodology, specifically the security evaluation process of IT outlined by the CEM. An ontological approach has been proposed to help coordinate this complex evaluation process. The previously existing ontology focused on the IT product development according to CC is expanded by considering evaluation issues. To express the IT security evaluation in accordance to CEM, ontology properties, classes, and individuals are elaborated. The use of ontology is demonstrated by a vulnerability study of a basic firewall. The paper emphasises the importance of expanding this ontology to include complete vulnerability analysis of various IT products and assurance levels. To assist the evaluator during the certification process, *Ekclhart (2007)* developed the CC

ontology tool based on an ontological representation of the CC catalogue. The CC ontology tool can help with tasks like preparing an evaluation process, reviewing relevant documents, and generating reports. The tool is aimed to reduce the amount of time and money required to complete a certification. *Sinnhofer et al. (2015)* have devised a scheme for selecting appropriate evaluation paradigm in accordance with CC in order to promote agile/modular design process for security certification in order to shorten the period between effective certification and the completion of product development. Moreover, since the most appropriate paradigm is selected, the costs of reevaluating the developed modules/product can be held as minimal as possible, optimising the reuse of previously tested modules, and allowing a direct incorporation of the evaluation facility in the process such that the feedback is directly applied to the next production iteration.

It is evident through the literature reviewed that existing work in the context of CC is largely focused to improve the CC certification and assurance procedure as well as the vulnerability assessment conducted as part of CC evaluation. Furthermore, some studies have discussed the significance of CC in software development life cycle. This means that existing research in the domain of CC depicts this standard as a criterion for improved and robust security controls. Therefore, efforts are made by different countries to become authorizing CC members. For this, a procedure to develop and certify a scheme, CB and evaluation facility must be followed, which is troublesome and majority of the developing countries find it extremely challenging to implement the whole procedure. Also, the reviewed literature clearly shows that research in this area is very limited. In such a scenario, there is a dire need of having a comprehensive survey that can act as a guideline for developing states to implement CC and attain maximum benefit by evaluating their products according to international standards and commercializing them globally. In addition, it can also assist consuming members of CC to become authorizing members and conduct evaluations.

## RESEARCH METHODOLOGY

The methodology employed for the purpose of carrying out the survey in this paper begins by searching keywords related to CC and its security evaluation procedure with regard to CC schemes on search engines like Google/Google Scholar. Research was then initiated from a pool of articles and most relevant sources were narrowed down to study further and comprehend founding understanding of CC. For the information regarding CC schemes of different countries, CC online portal was found to be the most comprehensive site encompassing all-inclusive CC reports and documents which was used to gather required information. Moreover, for a comprehensive survey, selection of countries with good cybersecurity efforts is an important factor. For this, *GCI (2020)* was consulted. Based on GCI ranking, CC schemes of four countries were selected. Among the top twenty most committed countries, US and UK comes first and second position with the GCI score of 100 and 99.54, respectively, whereas Singapore and Netherlands are at fourth and sixteenth position with GCI score of 98.52 and 97.05 respectively. An extensive study of important factors of evaluation and certification process of IT products was carried out to present a comprehensive survey of CC schemes of US, Singapore, UK and Netherlands.

| Table 1 Specifications of CC scheme by CCRA. | |
|---|---|
| **Entity** | **Roles & Responsibilities** |
| Evaluation Facility | Role based access control, requisite compliance with standards, withdrawal procedures, information sanitization, steps for evaluation, meeting, existence of security policies, evaluation facilities' tasks, legal agreements, existence of qualification levels, government/industry owned, conflict of interest, assurance continuity |
| Certification Body | Role, levying charges, interaction with evaluation facility, assurance continuity, notification of change, surveillance, publication responsibility, guidance for industry, withdrawal, consistency assurance, dispute handling, record sharing, validation, termination, evaluation steps, use of certification marks, requisite compliance |
| Certification Scheme | Management, structure, components, role of sponsor, participants, new interpretations, meeting, type of products, validation period, surveillance, nature of approved PPs, assurance continuity |
| Accreditation Body | Role of team, accreditation procedure, surveillance, role of accreditation |

Later on, based on the survey, a comparative analysis of the four schemes was performed to recommend some of the best practices to adopt while developing a CC scheme and to conduct CC evaluations.

## SURVEY OF COUNTRY LEVEL IMPLEMENTATION OF CC SCHEMES

In this section, the survey of country level schemes has been carried out. The requirements to develop a scheme as defined by CC are presented in Table 1. Basically, the schemes are responsible for managing the evaluation, validation, and certification procedure as per requirements given by the CCRA. The schemes perform and oversee the accreditation of evaluation facilities which are being actively involved in the evaluation and are accountable for issuing certificates in a particular region. Furthermore, another important responsibility of the schemes is to manage communication and disputes, which are raised among different bodies (*Dasso, 2016*; *Wooderson, 2017*). Apart from this, the scheme also maintains the list of validated products to ensure international recognition among CC members. To carry out evaluations and certifications in compliance with CC, seventeen certificate authorizing schemes have been developed and approved by CC. These schemes are responsible for managing the security evaluation process within their region (*Sohn, 2017*).

This research presents a comprehensive survey of country-level implementation of US, UK, Singapore, and Netherlands schemes in order to conduct security evaluation. As these countries are highly committed towards cybersecurity therefore, this research will definitely assist in identifying best practices that developing countries should follow to practice CC evaluation process efficiently. Following is the discussion of CC schemes along with their comparative analysis. The main participants of the schemes include but are not limited to sponsor, scheme, CCTL, accreditor and TC. Sponsor is an entity which requests and pays for the evaluation of COTS product and should cooperate with CCTL in terms of providing required technical materials (*NIAP, 2019a*, *SCCS, 2018b*,

| Table 2 Summary of accreditation body's tasks. | | | | |
|---|---|---|---|---|
| Parameters | UK Scheme | Netherlands Scheme | US Scheme | Singapore Scheme |
| Accreditation Body | UKAS | Netherlands Dutch Accreditation Counsel | NVLAP | SAC |
| Role of Accreditor | CLEF compliance validation with defined requirements. | Testing of ITSEF as an evaluator | Compliance approval/validation of lab with NIAP, NVLAP, NIST handbook 150, and NIST handbook 150-2 | Compliance validation of labs with SAC and SCCS |
| Accreditation Procedure | Usage of evaluation criteria, methodology, and CGOR | not specified | Usage of defined Procedures, NIST Handbook 150 and Information Technology Security Testing-Common Criteria, and NIST Handbook 150-20 | Validation of lab's compliance with ISO 17025, ISO 27001, ISO 27002, CEM, and SCCS and capacity to evaluate upto EAL 4 |
| Surveillance by Accreditation Body | After 4 years interval | not specified | Audit by NIAP and NVLAP | Audit activities by SCCS and SAC |

*UK-Scheme, 2013c*, *UK-Scheme, 2014b*). TC is governmental industrial partnership which is mainly responsible for developing PPs.

## United States scheme

**Introduction:** CCEVS or the NIAP is a scheme responsible for the security evaluation of COTS products in accordance with the CC. The validation body under the CCEVS is led by a Director and Deputy Director selected by NSA and NIST personnel (*NIAP, 2019a*). NIAP performs security assessments against the NIAP-approved PPs. The scheme consists of five important entities including laboratory director, approved signatories, authorized representative, evaluation team leaders, and senior evaluators.

**Accreditation procedure of the testing facility:** Organizations interested in performing evaluations and acquiring CCTL go through a series of steps. In the US, NVLAP is accountable for accrediting the testing facility and Table 2 presents its tabular form. Moreover, following is the list of NIAP and NVLAP requirements for acquiring CCTL approval.

1. **NIAP requirements:** NIAP verifies the specified requirements (*NIAP-Certificate, 2017*) by confirming the 'letter of conduct' being submitted by the applicant CCTL. If all requirements are fulfilled, then the status of NIAP-approved CCTL will be granted and documented.

2. **NVLAP accreditation:** It entails a candidate CCTL to show conformance with the methodological and technical criteria to conduct security evaluations of IT products (*NIAP-Certificate, 2017*). The assessment process consists of proficiency testing, an initial on-site visit and laboratory management system review. Important steps of the accreditation process are management system review (*NIAP-Certificate, 2017*), technical requirements for accreditation (*NIAP-Certificate, 2017*), proficiency testing, and NVLAP review.

**The flow of evaluation and certification of IT product**: The validation and evaluation of COTS along with the responsibilities of sponsors, CCTL, and the scheme in terms of evaluating IT products are discussed below. Once the sponsor has developed the ST and the strategy to share technical details, evaluation should be started (*NIAP, 2019b*, *Gossamer-Laboratories, 2019*; *NIAP, 2014*).

1. **Phase I:** First and the foremost step is to choose an appropriate CCTL. Another important step is to ensure that conflict of interest should not be present between evaluators and consultors (*Gossamer-Laboratories, 2019*).
2. **Phase II:** This phase begins with the check-in meeting with the sponsor and concludes with getting a final verdict (*NIAP, 2019b*). The responsibilities of evaluation facility with respect to NIAP are shown in Table 3.
3. **Phase III:** This phase comprises delivery of the evaluation document, PCL posting, and certificate issuance. In this phase, CCTL will provide ETR, ECR, VR, and TRRT to the validator. Once the review has been completed, the validator will complete VR. The NIAP director, in turn, will either prepare a CC certificate and forward it for signature or notify the sponsor and CCTL of the unsuccessful evaluation along with the rationale. In case of acceptance, mutually recognized CC partners will be notified (*Gossamer-Laboratories, 2019*; *NIAP-Certificate, 2017*).
4. **Phase IV:** Once the certificate has been issued, it is essential for the scheme to perform certificate monitoring and assurance continuity, which are discussed in the subsequent sub-subsections (*Costa, 2019*).

**Validation process:** It is the process of independent assurance that the evaluation has been conducted in accordance to NIAP policies and CCTL findings drafted in the ETR. CCTL will expedite the overall evaluation process by eradicating the requirement for the conventional VORs (*NIAP, 2014*; *Gossamer-Laboratories, 2019*; *Jacobs, 2015*; *Leaman, 2015*; *NIAP-Certificate, 2017*).

**Surveillance:** NIAP performs surveillance tasks in the following forms.

- **CB validator:** This process provides supervision at different milestones during evaluation. Important milestones include KO, IVOR, FVOR, TVOR, and evaluation conclusions (*Jacobs, 2015*; *Leaman, 2015*). Based on NIAP, Table 4 presents important responsibilities of CB.
- **Scheme and accreditation body:** It is essential for the validator to notify scheme management about the deficiencies on the part of CCTL. The roles and responsibilities performed by the NIAP is presented in Table 5.

**Communication management:** It can be done in the following ways:

- **Role of validator in managing communication:** The role of validator is extremely important in managing communication between the evaluation facility, CB, and scheme (*Jacobs, 2015*).

- **Validation of evaluation results:** Its role is significant in verifying and communicating evaluation results, records, and ETRs in terms of completeness and technical accuracy.
- **NIAP representative:** Another important responsibility of validator is to serve as a NIAP representative by providing a central point of contact between CCTL and NIAP. Moreover, another important role of the validator is to forward technical queries to suitable TRRT for comment and review.
- **CCTL support:** Validator provides support to CCTL by managing communication with the evaluation team when required and by ensuring that the team is aware of all suitable test methods and evaluation techniques. Moreover, it suggests information to be incorporated in records and ETRs for effective and efficient evaluation.

**Suspension or withdrawal of accreditation/approval:** If CCTL has not been organized with NVLAP and NIAP requirements then its status will be suspended or withdrawn. If CCTL status is 'withdrawn' then CCTL will cease all evaluation activities and the lab has to reapply for accreditation (*Jacobs, 2015*).

**Records management:** The major activities involved are records and procedures, orientation meeting for validators to get an idea about records' availability, storage, processing, and management. Moreover, to comply with the quality system of a scheme, it is imperative for the validators to keep the record of the work in an organized manner. To serve this purpose, records should contain a unique identifier at the top right in a standardized format. Furthermore, it is significant for validators to identify and protect the specified proprietary information. Official records should be closed out too and forwarded to the records manager within 30 days of final package delivery. In addition to that, the scheme will keep the evaluation record for at least 5 years.

**Assurance continuity:** Assurance continuity describes a method to minimize repetition in security evaluations. Maintenance is about the procedure, the developer practices to update documentation of a changed TOE whereas re-evaluation process assesses changed TOE (*Jacobs, 2015*). Due to the change, the CCTL/developer should submit an IAR for the products at least 30 days before the assurance maintenance. In case of major changes, CCTL must conduct tests and generate ETR. Finally, new VR, PCL, and certificate will be issued by CB. With respect to NIAP, Table 4 presents important responsibilities of CB. In case of minor changes, the developer fixes the bugs and the addendum will be made to the PCL along with documented fixes and patches.

**Dispute handling:** It is the responsibility of validator to respond to ECR inquiries and TRRT in timely fashion and scheme should support them (*Gossamer-Laboratories, 2019*).

**Interpretations:** To perform TOE evaluation, evaluation applicable in accordance with NIAP policy, CEM, and CC interpretations should be applied. The CCTL is accountable for the identification and application of appropriate interpretations.

## Singapore scheme

**Introduction:** SCCS is managed and owned by CB under the scope of CSA. The security evaluation of IT products is conducted by the approved CCTL, which is compliant with SCCS and accredited by SAC followed by the verification of results by the CB.

**Table 3 Summary of evaluation facility's tasks.**

| Parameters | UK Scheme | Netherlands Scheme | US Scheme | Singapore Scheme |
|---|---|---|---|---|
| Evaluation Facility | CLEF | ITSEF | CCTL | CCTL |
| Structure of Facility | 5 main components | not specified | not specified | not specified |
| Owner of Lab Facility | commercial or governmental | nil | Commercial, non-governmental | Third-party commercial |
| Requisite Compliance with Standard | UKAS, CGOR, and ISO/IEC-17025 | ISO/IEC-17025 | NIST Handbook 150, NIST Handbook 150-20, NIAP, and NVLAP | ISO 17025, SCCS, and SAC |
| Information's Sanitization | By doing output sanitization before sending it to sponsor | Achieved by secure ICT infrastructure | By Information Security Policy and Statement for Non-Disclosure of Proprietary Information | By securing information in ETRs, ORs, etc. |
| Availability of Security Manual | Yes | Yes | Yes | Policy conforming to ISO 27001/27002 |
| Qualification Levels of Evaluator | 3 levels, which are trainee; qualified, and specialist staff | not specified | not specified | Two members having practical knowledge of IT security |
| Legal Agreements | National or international agreements, full or provisional appointment, ownership rights, UKAS assessment, ETR publication, or between participants | Licensing and certification agreement | Policy for security and non-disclosure of information, Contract for documentation sharing, and agreement to share deliverables | Contract for conformance to the scheme-approved PPs and evaluation and certification agreement |
| Withdrawal by Evaluation Facility | At least 3 months prior notice to CB | not specified | In case of facility's non-compliance with NVLAP and NIAP, its status will be suspended or withdrawn | Sponsor can withdraw CCTL's appointment *via* CB-approved written notice |
| Records and Procedures Meetings | Meeting with the POC, CPR, Meeting minutes, and CCUKSG meetings | Kick-off meeting with CB | Meeting of CCTL with validator | TKM to discuss EWP, evaluation scope, and plan tasks, Task Close-down Meeting for the synopsis of evaluation tasks |
| Tasks of Evaluation Facility | Production of EWP, participation in TSR, review of ST, and validation of deliverables' availability | Agreement with sponsor before evaluation' initialization, CC compliance validation of TOE, production of ETR and EWP, and archiving of evaluation evidence | Implement scheme and CCRA policies, regulate information flow between lab and NIAP, production of ETR, ECR, VR, TRRT, OR, EWP, etc.,ST review, management of legal contracts | Perform secretarial functions to manage and record meetings, evaluation according to ISO 17025, and feasibility study to analyze the cost, time, and scope of evaluation, prepare SER, OR, and ETR, conduct EPM before and after the commencement of AVA and ATE and upon completion of ASE, provide test plan for AVA and ATE to CB, collect relevant configurations and documentation, compute IAR for assurance continuity, and handle dispute |

(Continued)

| Table 3 (continued) | | | | |
|---|---|---|---|---|
| Parameters | UK Scheme | Netherlands Scheme | US Scheme | Singapore Scheme |
| Conflict of Interest | No CLEF staff can be hired by sponsor within 2 years after his termination from CLEF | The evaluator cannot develop, evaluate, or advise about the TOE of his own company | Lab cannot provide evaluation and advisory services to the same TOE | CCTL should not provide consultancy to the same product. Moreover, the developer and evaluator can not be same. |
| Termination Agreement | not specified | ITSEF can terminate certification agreement by notifying the date | not specified | CB can revoke a certificate if any involved entity breaches the policy |

The management board of SCCS is the ultimate authority dealing with the evaluations being conducted and certificates being issued (*SCCS, 2018b*).

**Accreditation procedure of the testing facility:** Evaluation facility should be accredited by SAC or by other recognized body in compliance with ISO-17025. The lab must have a security policy compliant with ISO 27001 (*ISO, 2013b*) and ISO 27002 (*ISO, 2013a*). To gain accreditation, CCTL must be capable of performing evaluations at EAL-4 and be compliant with CEM and SCCS policies. Table 2 presents the procedures adopted by SCCS to accredit evaluation facility.

**The flow of evaluation and certification:** The evaluation will be conducted in the following phases (*SCCS, 2018c*).

- **Pre-evaluation phase:** The pre-evaluation phase involves feasibility study which states that if the sponsor wants to acquire a certification of an IT product, he will involve CCTL to perform the evaluation under legal contract by claiming conformance to a scheme-approved national PP, endorsed cPP or certified PP recognized by CSA. The products not claiming conformance with any of the above-mentioned conditions will be accepted up to EAL2+ (*SCCS, 2018b*). However, for certifying higher levels, the explicit written requirement by Singapore Government Agency is essential. Based on the evidence provided by sponsor, CCTL will conduct a feasibility study to analyze the cost, time, and scope of the evaluation. The major responsibilities of laboratory facility with respect to SCCS are discussed in Table 3.
  Application review is also included in this phase which states that once the application has been received, the CB will review the form, ST compliance with CC and updated EWP. If CB decides to conduct the evaluation, a 'Letter of Acceptance' will be provided to the sponsor and in turn, the confirmation will be granted to CCTL. A certifier will be assigned to supervise the overall evaluation (*SCCS, 2018c*). Based on the Singapore scheme, Table 4 presents important responsibilities of CB. After accepting the TOE for evaluation, CB will call kick-off meeting with the evaluation working group to confirm and discuss EWP to comprehend evaluation scope and address the applicability of the project's supporting documents. TKM is considered successful if all entities (CB, CCTL, sponsor) approve TKM meetings (*SCCS, 2018c*).

- **Evaluation phase:** Upon successful closure of TKM, CB will list particular IT product under evaluation and will perform assessment in accordance with the EWP. Test plan

for AVA and ATE should be provided to CB at least 2 weeks before EPM. At least, one EPM must be conducted after the completion of ASE (*Common-Criteria, 2017*) and one before and after the commencement of AVA and ATE (*Common-Criteria, 2017*). Table 4 presents important responsibilities of CB in accordance with SCCS. Moreover, CCTL must perform evaluation activities in accordance with ISO 17025 (*SCCS, 2018c*; *SCCS, 2018a*).

SER and ETR are the technical reports which are involved in evaluation phase. SER must be submitted to CCTL and CB upon completion of each evaluation task. SER contains verdicts of assessment activities from CCTL along with their justifications. The CB, in turn, reviews the SER and updates observations in SER-RR. For independent penetration and functional tests to be performed on TOE, CCTL should document test requirements, configurations, details of exploitable vulnerabilities, and scripts in SER. The major responsibilities of laboratory facility by considering SCCS are discussed in Table 3. ETR includes the information derived from ORs and SERs, which are then submitted to the CB. Vulnerability analysis must be conducted within 6 months prior to ETR submission otherwise CCTL should perform fresh vulnerability analysis. For international recognition, CCTL should be accredited to ISO-17025 before final ETR submission to the CB (*SCCS, 2018c*; *SCCS, 2018a*).

- **Conclusion phase:** Finally, Task Close-down Meeting will be arranged once the CB approves the ETR. During EPM, the CCTL will provide a brief synopsis of all evaluation tasks. The approved ETR will provide a baseline for preparing CR, which contains the final verdict regarding successful/unsuccessful evaluation.

- **Certificate awarding phase:** Once the evaluation has been completed, the process of awarding certificate begins. The CB will prepare a report containing forms, ST, ETR, CR, and CC certificate. Upon approval by CB's head, the certificate with developer name will be available on CPL for international recognition. This certificate will be valid for 5 years, which can be extended through assurance continuity (*SCCS, 2018a*).

**Surveillance:** The identification of roles and responsibilities taken into account by the SCCS are presented in Table 5. This scheme performs the surveillance of CCTL either through CB and accreditation body. tasks in the following ways (*SCCS, 2018a*).

**Communication management:** Different entities as mentioned below have been involved in managing communication which are; NSC, CCTL, and CB (*SCCS, 2018c*).

**Certificate withdrawal, revocation, and termination:** The fee will not be refunded in case of termination, withdrawal or suspension of a certification procedure. Upon certificate revocation, CCTL will immediately cease the usage of CC certificate and will add details of the TOE to HPL. The CB can terminate the on-going certification process without certificate issuance by giving notice if the specified conditions are met (*SCCS, 2018a*). Sponsor has an authority to replace or withdraw the CCTL's appointment by serving a written notice. The CB is entitled to accept/reject the application. The project details will then be moved to HPL and will be marked as 'withdrawn'.

**Table 4 Summary of certification body's tasks.**

| Parameters | UK Scheme | Netherlands Scheme | US Scheme | Singapore Scheme |
|---|---|---|---|---|
| CB | CESG CB | TÜV Rheinland Nederland CB | Validation body is led by NSA and NIST selected Director and Deputy Director | CB under scope of Cyber Security Agency of Singapore (CSA) |
| Role of CB | not specified | Assure the NSCIB's maintenance by considering appeals lodged by third parties, issue certificates, and provide licenses to ITSEFs | Ensure policies' enforcement, approve/publicize/monitor CCTLs, secure information, develop PPs and their compliance with ETRs, promote CC certificates and logos' integrity. | Manage scheme operations, oversight evaluation, issue CR and certificate, and accredit facility. |
| Charges levied by CB | Evaluation, reevaluation, assurance maintenance, training, and UKAS charges | Certification investigation cost, penalty for breaching conditions, certification mark, scheme setup, and maintenance charges | Certification and lab accreditation cost | Lab accreditation and product certification charges |
| Legal Agreements | Agreement between entities for non-disclosure, etc. | Licensing agreement | Evaluation, licensing, and non-disclosure agreements | Evaluation contract between sponsor and CB |
| Assurance Continuity Management | For modified TOE, the sponsor needs to apply for assurance continuity | With assurance continuity, modified TOE can be certified without formal reevaluation | Evaluations of modified products should be dealt properly | For modified TOE, previous evaluation should be fully utilized |
| Dispute Handling | Dispute with CLEF can be resolved by CB head or scheme senior executive | Dispute with ITSEF can be resolved by Netherlands' courts in Hague | NIAP should handle the disputes while ensuring the interests of all stakeholders. | Disputes are resolved through SCCS's Complaints, Disputes and Appeals process |
| Interaction of CB with lab | Evaluation Progress Meeting (EPM), CPR, and staff changes must be notified | Meetings regarding lab's scope, certification, accreditation, and suspension | Verification of evaluation results, records, ETRs, and policies, and management of queries. | For conducting evaluations and communicating staff changes |
| Surveillance by CB | CB monitors the CLEF's performance on annual basis | not specified | NIAP validates CCTL performance on regular basis | CB validates lab performance |
| Withdrawal by CB | Withdrawal from CLEF by giving 3 months prior notice | CB can reject the CCTL license | Withdrawal from CCTL on non-compliance with the policy | Withdrawal from CCTL by giving 1 month written notice to CSA |
| Record Sharing Means | Email policy is defined for this purpose | Certified email or certified document | Electronic form through e-mail by using identifier | not specified |
| Validation Procedure | not specified | not specified | Check-in, evaluation, and approval package phase | Review of SER Review Report (RR) and preparation of CR by CB |
| Requisite compliance with standard | not specified | Kwaliteits-managements-systeem | ISO-17025, NIAP | SCCS and CSA |

| Parameters | UK Scheme | Netherlands Scheme | US Scheme | Singapore Scheme |
|---|---|---|---|---|
| Termination Agreement | CLEF's termination in case of non-compliance | Facility's termination due to inappropriate evaluations by issuing 3 months prior notice | not specified | Sponsor or facility's termination in case of inappropriate actions like certificate misuse, etc. and inaccurate evaluations respectively |
| Usage of International Certification Marks | not specified | A joint claim will be signed with CB | NIAP, CSEG and CC marks | SCCS and CB marks |
| Steps for Evaluation | not specified | not specified | Check-in, evaluation, and approval package phase | Communication with the sponsor and facility, review of EWP, and preparation of CR |
| Records and Procedures Meetings | CCUKSG, TSM, TSR | Kick-off meeting with ITSEF | Records and procedure oriented meetings | Kick-off meeting, EPM, and close down meeting |

**Records management:** The scheme does the records management by sharing records *via* TKM and EPM meetings, publish ST after sanitizing the sensitive information, and sanitize reports if developer and sponsor are different entities.

**Assurance continuity:** AC can be of different nature including re-evaluation and certificate maintenance. When the impacts of changes made to the TOE on assurance baseline are minor, then certificate maintenance will be performed (*SCCS, 2018a*). Re-evaluation will be applicable if such effects are major. The sponsor will exercise impact analysis procedure and draft IAR to determine whether these impacts are major or minor. Based on it, CB will decide AC. Based on the SCCS, Table 4 presents important responsibilities of CB.

**Dispute handling:** Disputes are resolved through SCCS's Complaints, Disputes and Appeals process (*SCCS, 2018c*; *SCCS, 2018a*).

**Use of logos and protective marks:** The sponsor can use SCCS and CC marks provided that the evaluation has been successful. These marks should be used in standard form except the alterations in monochromatic color schemes and size (*SCCS, 2018a*).

## Netherlands scheme

**Introduction:** The objective of NSCIB is to certify the IT products in accordance with CC, which is composed of ISO/IEC 15408 and ISO/IEC 18045 (*NSCIB, 2017*).

**Accreditation Procedure of the Testing Facility:** The procedure for licensing the ITSEF with the CB consists of the following phases (*NSCIB, 2017*):

- **Start of Licensing Process:** Firstly, the ITSEF needs to submit an application to the CB. In case of rejection of the application by the CB, the notification should be sent to the ITSEF which should counter all the reasons to get the application accepted. In the case of acceptance, the notification informing the licensing-ID and the assigned certifier must be sent to the ITSEF.

- **Kick-off Meeting:** The kick-off meeting after the acceptance of the application will be arranged in which ITSEF's knowledge regarding the licensing process is ensured by the CB, the status of ITSEF with respect to ISO/IEC-17025, evaluation of testing, and verification of the preliminary licensing work plan and draft of licensing agreement is formulated.

- **Licensing Agreement:** The licensing agreement is sent by the CB to the ITSEF and in case, the CB wishes to reject the licensing agreement, it can only reject within the period of 30 days after the letter is sent to the ITSEF. Only those ITSEFs are authorized to perform evaluations which are licensed by the CB (*NSCIB, 2017*). Table 2 presents the procedures adopted by NSCIB to accredit evaluation facility.

- **Execution:** The licensing plan can be subsequently executed by the ITSEF and the CB.

**The flow of evaluation and certification:** In the evaluation facility ITSEF, when the TOE or PP is evaluated for the first time, the process goes through three phases (*NSCIB, 2017*; *Weith, 2014*). First one is the preparation phase, which involves the submission and processing of the formal application. The major responsibilities of laboratory facility by considering NSCIB are discussed in Table 3. Second one is the monitoring phase, which involves the evaluation of product under the control of CB and results in the production of ETR. Table 4 presents important responsibilities of CB in accordance with NSCIB. Finally the certification phase incorporates the final actions which are needed to be taken and the generation of the certificate.

**Surveillance:** The scheme is responsible for performing surveillance activities. The NSCIB does not conduct an audit periodically after the issuance of certificate except for the reception of complaint (*NSCIB, 2017*). The identification of roles and responsibilities performed by the NSCIB is presented in Table 3.

**Assurance continuity:** In case of changed TOE, the sponsor is required to apply for re-certification or maintenance of assurance depending upon the nature of changes caused.

**Dispute handling:** If some dispute regarding the certification agreement cannot be resolved between the concerned parties then that can be settled according to the laws of Netherlands courts in the Hague.

**Interpretations:** If the assessment guidelines during evaluation demand clarity then the ITSEF, CB, and the sponsor shall find criterion interpretation which is acceptable but if no acceptable interpretation is found, the ITSEF can submit RI at CB. Moreover, if CB is unable to answer the RI, then it can make a request to other international groups of experts, ITSEFs, or CBs (*NSCIB, 2017*).

## UK scheme

**Introduction:** For COTS products' evaluation, the UK IT Security Evaluation and Certification Scheme is used which is operated by the CESG. Its CB is CESG CB, which is developed by the scheme (*UK-Scheme, 2013c*). The CESG CB is responsible for appointing the CLEFs to carry out products' evaluations (*UK-Scheme, 2013c*; *UK-Scheme, 2014b*). The UK IT Security Evaluation and Certification Scheme is comprised of the SINs, IICC, UKI, UK Scheme Publications, assurance maintenance documentation, etc.

**Table 5 Summary of certification scheme's tasks.**

| Parameters | UK Scheme | Netherlands Scheme | US Scheme | Singapore Scheme |
|---|---|---|---|---|
| Certification Scheme | UK IT Security Evaluation and Certification Scheme | Netherlands Scheme for Certification in the Area of IT Security (NSCIB) | CCEVS/NIAP | SCCS |
| Management of Scheme | NSCIB is operated by the UK's National Technical Authority for Information Assurance | SOGIS-MRA and NLNCSA | NIST and NSA | CB under the scope of CSA |
| Components of Scheme | SINs, CC documents, IICC, UKI, Scheme Publications, and Assurance Maintenance Documents | International CCRA and European SOGIS mutual recognition arrangement | International CCRA, NIAP, NIST and NSA | International CCRA and CSA |
| Participants of Scheme | Sponsor, developers, vendor, procurement body, and accreditor | Sponsor and developers | Sponsor, NIAP, CCTL, TC | Sponsor, developer, CB, CCTL, consultant, evaluation working group |
| Role of Sponsor | Support supply for TOE certification and assurance continuity | Payment of all dues and communication with CB and ITSEF | Inform about modified TOE, answer queries, provide access for evaluation, and clarify scope of ST information to be made public | Timely submission of the deliverables for evaluation |
| Meetings Related to National or International Issues | CCUKSG meeting is held after every 3 to 4 months | not specified | not specified | not specified |
| Management of New Interpretations | For new interpretation, the issue raised by facility, is finally considered by CCUKSG | The Request for new/CC Interpretation (RI) submitted by the ITSEF will be discussed with an international experts to add the resultant interpretation in the scheme | The CCTL identifies and apply validator-verified interpretations with the help of Common Criteria Maintenance Board (CCMB), NIAP policy statements, etc. | not specified |
| Certificate Validation Period | nil | Product's certificate validity period is 5 years and certificate validity of facility is 2 years | 5 years | 5 years |
| Types of Evaluated Products | Integrated Circuits (ICs), Smart card-related systems, Network-related systems, Operating Systems, and other systems | Boundary and data protection systems, ICs, smart card-related systems, Key management systems, Network-related systems, and other systems | Access control systems, Biometric systems, Boundary protection systems, Data protection, Databases, Detection systems, ICs, smart card-related systems, Mobility, Multifunction devices, Key management systems, Operating systems, networking-related systems, Products for digital signatures, other systems. | Network and network related devices and systems |

(Continued)

| Table 5 (continued) | | | | |
|---|---|---|---|---|
| Parameters | UK Scheme | Netherlands Scheme | US Scheme | Singapore Scheme |
| Nationally or Internationally Approved PPs | not specified | not specified | NIAP performs evaluation against the NIAP-approved PPs. | Use of scheme-approved national PP, endorsed cPP or CSA-recognized PP is preferred. In other cases, evaluation can be accepted up to EAL2+ but for higher levels, written notice by Singapore Government Agency is essential. |
| Scheme structure | not specified | not specified | 5 main components; Laboratory Director, Authorized Representative, Approved Signatories, evaluation team leaders, and senior evaluators | not specified |

**Accreditation Procedure of Testing Facility:** For the two categories, the CLEF will be evaluated to give UKAS accreditation (*UK-Scheme, 2013b*; *UK-Scheme, 2014a*). First one is the permanent lab and second one is the on-site testing lab. The CLEF is needed to perform the evaluations and their respective reports in accordance with the UKAS accreditation. Table 2 presents the procedures adopted by UK scheme to accredit evaluation facility. The scope of UKAS accreditation includes use of evaluation criteria, evaluation methodology, and use of CESG Generic Test Method.

**The Flow of Evaluation and Certification:** The certification procedure is decomposed into the following stages (*UK-Scheme, 2014a*).

- **Preparation:** In order to determine the suitability for the evaluation of TOE, relevant information is gathered. The major responsibilities of laboratory facility given in UK scheme are discussed in Table 3.
- **Evaluation and Certification Phase:** Based on UK scheme, Table 4 presents important responsibilities of CB (*UK-Scheme, 2016*; UK-Scheme, 2013b).
- **Assurance Maintenance Phase:** Assurance maintenance phase is essential to have confidence that the modified TOE is secure without the formal re-evaluation.
- **CWP:** For outlining the certification work which needs to be carried out by the CESG CB, the CWP is used which can later be modified too. Certification activity can be review of scope of TOE, etc. (*UK-Scheme, 2014a*).

**Surveillance:** The surveillance activities are performed by accreditation body and CB. Initially, the assessment visits are carried out by the UKAS assessors. After the date of accreditation, the first surveillance visit is usually carried out after the duration of 6 months and successive visits are conducted on yearly basis. Moreover, in order to update the accreditation of an existing CLEF, extended surveillance visits are carried out after which the agreement is signed between the UKAS and CESG CB to decide the scope of extended accreditation schedule (*UK-Scheme, 2014a*; *UK-Scheme, 2013a*; *TSM, 2013*).

Following the UKAS reassessment, the CESG CB will carry out the surveillance by showing involvement in the procedure for certification of evaluations.

**Suspension or Withdrawal of Appointment:** In case of the lapse of UKAS accreditation or if the CLEF breaches the appointment's conditions, the CESG CB can give short notice to withdraw the appointment of CLEF. If the CLEF wishes to withdraw from the scheme, it must give at least 3 months prior notice and the same notice period is expected from the CESG CB if it wishes to withdraw or renew the terms of appointment of CLEF. At CLEF appointment's termination, the CESG CB will determine whether any kind of evaluation work will be continued to let CLEF fulfill its contractual duties to its sponsors. If the scheme is to be terminated, then CESG CB can also withdraw all CLEF appointments on 6 months notice (*UK-Scheme, 2013b*; *UK-Scheme, 2014a*).

**Assurance Continuity Management:** The judgement of assurance maintenance for the changed TOE is based upon the IAR produced by the sponsor/developer to demonstrate the impact after changes to the TOE have been done, whereas, for the certified IT product, the assurance maintenance is assessed on the basis of ETR, CC certificate, ST, and CR (*UK-Scheme, 2013b*).

**Dispute handling:** If the dispute occurs between the CESG CB and the CLEF, then it can be resolved by the help of head of CB or the scheme senior executive.

**Interpretations:** The CCUKSG meeting is required to be held after every 3 to 4 months approximately (*TSM, 2013*, *EPM, 2013*; *CWP, 2013*). The CCUKSG meeting is focused on the discussion and agreement on UK issues, international issues, and the UK interpretation like security characteristics, CC portal, CESG website, CCRA, CC/CEM, SOGIS-MRA, ISO, cPPs, and other supporting documents.

## COMPARATIVE ANALYSIS

This section compares and contrasts the aforementioned four schemes to suggest some best practices that developing countries must follow while developing a scheme. This research evaluates certification scheme, evaluation facility, CB, and accreditation body of UK, Netherlands, US, and Singapore based on some parameters, which are discussed below in detail. Table 6 present this comparison briefly.

### Maturity and technical skills

US gained signatory status back in 1998, whereas UK and Netherlands started evaluating IT products in 2000. Singapore just attained signatory status and is assessing IT products at a great pace. By considering this factor, it can be inferred that the US, UK, and Netherlands are quite mature and can conduct evaluations with great technical capabilities. Similarly, the US has developed nine evaluation facilities, which are working in different domains to evaluate a diverse range of IT products. On the other hand, the UK and Netherlands have succeeded in developing two laboratories only. So far, Singapore is working efficiently and has already developed 4 evaluation facilities concerned in evaluating network and digital signature devices.

Another important factor to consider is the number of evaluated products which demonstrates the technical capabilities of a particular scheme. Till now, the US, UK,

**Table 6 Comparative analysis of CC schemes.**

| Parameters | Sub-parameters | UK Scheme | Netherlands Scheme | US Scheme | Singapore Scheme |
|---|---|---|---|---|---|
| Signatory status | Timeline | 1998 | 2000 | 1998 | January 2019 |
| No. of testing facilities | – | 2 | 2 | 9 | 4 |
| Maximum evaluated assurance level | – | EAL5+ | EAL7+ | Not specified | EAL4 |
| No. of evaluated products | – | 135 | 85 | 922 | 4 |
| No. of developed/certified PPs | – | 22 | 4 | 294 | nil |
| Lead responsible authority | – | CESG | Interior and Kingdom Relations Ministry | NSA/NIAP | Infocomm Development Authority of Singapore |
| Evaluation facility | – | CLEF | ITSEF | CCTL | CCTL |
| Certification Body | – | CESG CB | TV Rheinland Nederland CB | Validation body is led by NIST and NSA-selected Director and Deputy Director | CB under the scope of CSA |
| Accreditation body | – | UKAS | Netherlands Dutch Accreditation Counsel | NVLAP | SAC |
| Surveillance | Monitoring | Proper policy is defined with pe-riodic reviews | Policy is not specified | Partially defined, audit by accreditors | Partially defined, audit by SCCS and SAC |
| Strategic guidance | Guidance documents | Security manual, technical re-ports, meeting minutes | Security manual, technical reports, meeting min-utes | Security manual, technical reports, meeting minutes, policy documentation, NIST hand-book | Policy documents conforming to ISO 27001/27002, technical reports, meeting minutes |
| Capacity building | Trainings | Not specified | Not specified | Training, workshops and conferences | Hosting a conference |
| Structure | Scheme structure | Not specified | Not specified | Properly described | Not specified |
| Standardization | Standardized bodies and standards | UKAS, CGOR, and ISO/IEC-17025 | ISO/IEC-17025 | NIST Handbook 150, NIST Handbook 150-20, NIAP, and NVLAP | SCCS and SAC |
| Legal obligations | Agreements | Limited number of agreements | Licensing agreement and certification agreement | Several national agreements and policies | Some national agreements |
| | Requisite compliance | UKAS, CGOR, and ISO/IEC-17025 | ISO/IEC-17025 | NIST Handbook 150, NIST Handbook 150-20, NIAP, and NVLA | SCCS, SAC, ISO 17025 conformance |

| Parameters | Sub-parameters | UK Scheme | Netherlands Scheme | US Scheme | Singapore Scheme |
|---|---|---|---|---|---|
| Technical competency | Technical skillset with re-spect to evaluated products | Access control devices, ICs and smart card related systems, Net-work related devices, Bound-ary and data protection devices, Databases, key management de-vices, OS, Other systems | Boundary protection devices, ICs and smart card related devices, Digital signatures devices, Other systems, Access control devices, Data protection, Mobility, Network related devices, Operating Systems, and Trusted computing | Access control devices, Data and boundary protection systems, Mobility, Multi-function devices, Network related devices, Other sys-tems, Databases, Detection systems, Key management, OS | Network devices, Digital signatures, and Boundary and data protection devices |
| | No. of evaluated products with respect to device cate-gory | Access control devices (2), ICs and smart card related devices (31), Network related devices (23), Boundary protection de-vices (30), Data protection de-vices (1), Databases (13), key management devices (4), OS (12), Other system(7) | Boundary protection devices (4), ICs and smart card related devices (101), Digital signatures de-vices (15), Other systems (4), OS(2), Network related devices (17), Mobility (1), Data protection (1), Access control devices (1), Trusted computing (1) | Access control devices (37), Data protection devices (10), Mobility (20), Multi-function devices (3), Network related devices (51), Other systems (50), Boundary protection de-vices (3), Databases (25), Key management (12), Multi-function devices (3), OS (3) | Boundary protection devices(2), Data protection (2), Network devices (1), Digital signatures (2) |
| | Levels of evaluators | Properly defined | Not defined | Not defined | Partially defined |
| | Policy regarding use of nationally developed/ approve d PPs | Not specified | Not specified | Yes | Both national and inter-national PPs can be used after CSA special ap-proval |
| | Assurance continuity | Properly defined | Properly defined | Properly defined | Properly defined |
| Comprehensiveness | Tasks of evaluation facility | Partially defined with some tech-nical details | Partially defined | Well-organized and properly elaborated | Defined |
| | Scheme | Partially described | Partially described | Well-organized and properly elaborated | Partially discussed |
| | Role of sponsor | Partially described | Partially described | Well-organized and properly elaborated | Properly discussed |
| | Evaluation steps | Not specified | Not specified | Partially defined | Well elaborated |
| | Validation procedure | Not specified | Not specified | Partially defined | Well elaborated |
| | Role of CB | Not specified | Properly described | Properly discussed | Partially described |
| | Role of sponsor | Properly described | Properly described | Properly discussed | Properly described |
| | Role of Accreditor | Specified | Specified | Detailed | Detailed |
| | Accreditation procedure | Partially defined | Not specified | Properly defined | Properly defined |

(Continued)

 

| Table 6 (continued) | | | | | |
|---|---|---|---|---|---|
| Parameters | Sub-parameters | UK Scheme | Netherlands Scheme | US Scheme | Singapore Scheme |
| Communication | Meetings regarding national/ international issues | CCUKSG meeting after every 3 to 4 months | Not specified | Not specified | Not specified |
| | Management of new interpretations | CCUKSG | International group of experts | Common Criteria Maintenance Board (CCMB) and NIAP | Not specified |

Netherlands, and Singapore have acquired first, second, third, and fourth positions respectively.

The technical competency of the scheme can also be evaluated by considering the maximum level of the product being evaluated. Netherlands, UK, and Singapore schemes possess technical skills of evaluating the products up to EAL7, EAL5, and EAL4 respectively. Another important factor under consideration is the number of developed PPs that can be used for evaluation. So far, the US, UK, and Netherlands have developed 294, 22, and 4 PPs respectively. In the US, NSA/NIAP are the lead responsible authorities that must be consulted in case of any query, whereas in the UK CESG is the responsible body. Netherlands and Singapore have nominated the Ministry of the Interior and Kingdom Relations and Infocomm Development Authority of Singapore as lead responsible bodies.

## Comprehensiveness

It is concerned with the completeness of evaluation procedures and policies. It is about how well a scheme is organized and discusses details of the followed procedures.

In a UK scheme, the tasks of the evaluation facility have not been defined in detail, whereas the US scheme elaborates the tasks of the evaluation facility properly. The significant tasks to be followed are to: implement the scheme, manage legal agreements, regulate information flow, prepare technical reports, and follow phases which assist in evaluation and certification procedure. The Singapore scheme describes all legal policies, procedures, agreements, and feasibility study with regard to the evaluation facility and the necessary guidelines in the preparation of technical reports and the management of records and meetings. In contrast, Netherlands scheme only provides guidelines regarding the management of agreements and the production of technical reports. As far as the documentation guidance or the components of scheme is concerned, the UK and US schemes provide a large variety of documents including the defined national and international standards, whereas Netherlands and Singapore schemes provide limited documents. Moreover, the UK, Singapore, and US schemes involve parties like the procurement body, consultant, and TC as additional participants, whereas Netherlands scheme limits the participants' role to sponsor and developers only.

The UK and Netherlands schemes do not specify any steps necessary for performing the evaluation and validation of an identified product whereas the US and Singapore schemes

do specify. According to the US scheme, the check-in, evaluation, and approval phases refer to the steps for evaluation which depict the submission of the TOE, testing of the TOE, and final declaration regarding TOE's status, may it be validation or non-validation. The description of steps for evaluation has been done by the Singapore scheme in detail, as it demands the need for submission of CAF to the CB. In case of acceptance, the CB sends an approval letter to the sponsor and the evaluation facility too. Immediately after that, a meeting is called by the CB which demands the need for participation from the evaluation facility to carry out reviews and generate CR.

Finally, coming over to the accreditation procedure, Netherlands scheme does not explicitly define the accreditation procedure but the UK scheme incorporates the usage of evaluation criteria, a methodology for evaluation, and CGOR for performing accreditation procedure. Moreover, the US scheme defines the accreditation procedure to be comprised of the guidelines provided by defined national standards and proficiency testing involving the initial on-site visit and review of the evaluation facility's management system. On the other hand, the Singapore scheme demands the need from the evaluation facility to be capable of evaluations at EAL4 and the facility must be compliant with defined national and international standards.

## Role of sponsor

The role of sponsor according to UK scheme demands the need to get all types of supporting documentation related to the TOE from him/her for the certification of the TOE and its assurance continuity, whereas Netherlands scheme applies the responsibility for the payment of all dues regarding the evaluation of TOE on the sponsor. Moreover, the sponsor also maintains the communication channel between the CB and the evaluation facility. The US scheme lays greater responsibility on the sponsor as he must inform if any modifications have been made to the TOE, provide access to the testing environment, clarify the scope of sensitive information of the respective ST, and answer all queries from the evaluation facility while the Singapore scheme explains the role of sponsor very briefly.

## Role of CB

In defining the role of CB, the UK scheme has not played any effective role while Netherlands, US, and Singapore schemes have explicitly discussed the role of CB in detail. According to Netherlands scheme, the CB should assure the maintenance of the quality system of evaluation facility, assist the body which issues certificates in making evaluation assessments, make a decision regarding issues of CB with the third parties, and perform issuance of certificates and licenses to the sponsors and evaluation facility respectively. The US scheme discusses the role of CB by implying the responsibilities like, the smooth functioning of evaluation facility, approval, publication, and monitoring of evaluation facility. Moreover, according to the US scheme, CB should perform information sanitization, development and validation of PPs, verification of ETRs' compliance with the developed PPs, promotion of the CC certificates, and assurance of the correct usage of scheme and logos of CC. In contrast, the Singapore scheme restricts the CB to manage the operations of scheme, deal with the certification application, supervise evaluation done

by the evaluation facility, issue CR and certificate, and play role in accrediting the evaluation lab.

## Role of accreditor

As far as the role of the accreditor is concerned, all the four schemes have defined it briefly. The UK scheme expects from the accreditor to check the compliance of the evaluation facility with the necessary accreditation requirements, whereas Netherlands scheme implies the responsibility of the accreditor to check the competencies of the evaluation facility as a tester as well as an evaluator. In contrast, as per the US scheme, the accreditor must approve and validate the compliance of the evaluation facility with the defined national standards. Similarly, the Singapore scheme validates the compliance level of the evaluation facility with the specified national standards.

## Technical competency

It deals with technical capabilities or skillset that the scheme possesses to carry out security evaluations and their management.

The qualification levels and the skillset of evaluators have been explicitly described by the UK and Singapore schemes. The UK scheme defines three levels of evaluators which are; trainee, qualified, and specialist staff to evaluate four different categories of products. Similarly, the Singapore scheme describes its evaluators to be two staff members having practical experience, which is presently focused on the category of network devices. In contrast, Netherlands and US schemes do not define the qualification level of evaluators rather these schemes only define the number of categories their evaluators can evaluate which are, six and eleven in the case of Netherlands and US schemes respectively. The comparison is presented in Table 6.

With respect to the types of products that are being evaluated, all four schemes have different expertise based on which they evaluate different categories of products. Those list of categories with respect to the UK, Netherlands, US, and Singapore schemes can be seen in Table 5.

As far as the acceptance of PPs by the schemes are concerned, the UK and Netherlands schemes do not specify any restriction but the US scheme approves the national PPs only. Similarly, the Singapore scheme also approves the national PP, endorsed cPP, and any other CSA recognized PP. Moreover, if the TOE is not based on the aforementioned PPs, then the Singapore scheme allows evaluation up to EAL2+ but other than that, the approval by Singapore Government Agency is mandatory.

## Assurance continuity

With regard to assurance continuity, the UK, Netherlands, and US schemes bound the sponsor to apply for it, in case the modifications have been made to the TOE after certification. The Singapore scheme augments the assurance continuity procedure by suggesting the use of previous evaluation work carried out to be reused in re-evaluation as it will provide assistance in case of the modified TOE.

## Legal obligations

This covers legal frameworks that the entities should comply with. The legal agreements and the requisite compliance with standards vary with respect to the chosen four schemes. The UK scheme enlists the legal obligations in the form of agreements related to full or provisional appointment, national or international issues, ownership rights, UKAS assessment during the evaluation, the release of ETR in future, participants involved in the evaluation, and compliance with the UKAS, CGOR, and ISO/IEC-17025. Similarly, Netherlands scheme also describes legal agreements related to licensing, certification, and compliance with ISO/IEC-17025. Furthermore, the US scheme describes security policy, agreement for non-disclosure of proprietary information, a contract regarding documentation sharing and conformance with defined standards, and the agreement regarding the sharing of deliverables. Just like the US scheme, the Singapore scheme restricts the evaluation facility to ensure conformance to the nationally approved PPs, SAC, and SCCS, and to sign the contract for evaluation and certification, in the form of legal obligation.

## Communication and engagement

To conduct evaluation successfully, communication and coordination between major entities are extremely important.

The communication and coordination of activities with respect to the UK scheme involve meeting with the POC, CCUKSG which focus on UK /international issues and national interpretations, CPR, and meeting minutes related to the EPM. In contrast, Netherlands scheme restricts the coordination activities to a kick-off meeting. Similarly, the US scheme describes the meeting of the validator with the evaluation facility so that he can understand mechanisms ensured by the facility for evaluation and record keeping. In the same way, the Singapore scheme describes task close-down meetings in which the evaluation facility provides a synopsis of evaluation tasks and the arrangement of TKM.

## Surveillance

It is important to oversee evaluation tasks, legal agreements, communication, technical reports, compliance and information flow for ensuring quality and credibility of CC assessment.

The UK scheme defines surveillance in terms of monitoring of the CB and the evaluation facility by the certification scheme. Similarly, US and Singapore schemes are responsible for ensuring a smooth process in evaluation and accreditation. The surveillance by the accreditation body in Netherlands scheme has not been taken into account. In contrast, the UK scheme defines the first surveillance by the accreditation body to be carried out after a period of 6 months, whereas the successive surveillance visits are to be conducted on a yearly basis and the full assessment is to be carried out after 3.5 years with respect to the accreditation date and thereafter 4 years interval. Similarly, the US and Singapore schemes describe the procedure of audit to be conducted by NIAP and NVLAP and by SCCS and SAC respectively.

## Transparency

The devised policies should be transparent and stringent in nature for ensuring accurate implementation of CC.

For ensuring transparency, the UK scheme defines information sanitization policy, withdrawal policy that implies condition on the evaluation facility to notify CB at least 3 months before carrying out withdrawal of appointment, conflict of interest policy so that the sponsor can not hire staff from CLEF within 2 years after his termination from the evaluation facility, but the scheme does not specify termination policy. Similarly, Netherlands and the US schemes define the sanitization of output by the implementation of secure ICT infrastructure, conflict of interest policy which in case of Netherlands scheme, deals with the clause that the evaluator can not develop, evaluate, or provide consultancy to the TOE, and in case of US scheme deals with the clause that the facility can not provide consultancy to the product. Unlike the UK scheme, Netherlands scheme provides termination policy too by mentioning that the evaluation facility can terminate the agreement with the CB by notifying the date prior to termination but in contrast, the US scheme does not specify the termination policy.

The Singapore scheme also defines transparency in terms of information sanitization policy and conflict of interest policy. Moreover, the sponsor can withdraw an appointment which needs to be approved by CB. The termination policy in Singapore scheme has also been discussed in detail which states that a certificate can be revoked if the developer or sponsor or evaluation facility breaches the terms and conditions, fails to disclose vulnerabilities and corrective actions, the evaluation activity has not been conducted for more than 60 consecutive days, the sponsor or the facility does not take actions in time, the evaluation scope has been changed without informing the scheme and CB, or the quality of the work performed by the facility is not up to the mark.

## Validity period

As far as the validity period of the certificate is concerned, the UK scheme does not specify any period of time. In contrast, the US and the Singapore schemes define the period of validity to be 5 years. The US scheme also defines the validity period to be 5 years and additionally specifies the validity of the site's certificate to be 2 years.

## Cost

With regard to the charges levied by the CB, the UK scheme states that the charges for evaluation, reevaluation, assurance maintenance, training course, and charges for UKAS assistance must be charged. In contrast, according to Netherlands scheme, certificate investigation cost, advanced invoice, breach of conditions' penalty, certificate cost, certification mark usage cost, and the setting up along with the maintenance of scheme cost must also be charged. The US scheme demands the cost of certification and accreditation of the facility's cost. Similarly, the Singapore scheme charges the cost of accrediting the facility and for doing certification of TOE with respect to the assurance level.

## Withdrawal procedure

The withdrawal by the CB is also defined explicitly by all four schemes. According to the UK scheme, the CB can withdraw from the facility by giving 3 months prior notice. In case the scheme gets terminated, then CB has the right to terminate the appointments of all facilities on 6 months' notice. Netherlands scheme states that CB can reject the application for the licensing process and will accept the renewed application if all the rejection reasons are fairly justified by the sponsor. Like the UK scheme, the CB can withdraw the appointment of facility based on defined policies. Similarly, the Singapore scheme can withdraw the appointment of the facility too by giving 1 month written notice to CSA.

## Certification

The usage of international certification marks in the UK scheme has not been specified. In contrast, Netherlands scheme lays down the condition to sign a joint claim with the CB and the sponsor if the latter wishes to use certification marks. Moreover, the CB must also ensure only authorized use of certification marks. The US and the Singapore scheme allow the use of their respective marks based on the defined policies.

## Termination procedure

The termination agreement in the case of the UK scheme can be signed between the CB and the facility if the latter does not comply with the defined rules. Netherlands scheme can terminate the certification process prematurely if the sponsor does not, meet the plan regularly, pay dues, or provide correct reports. Moreover, if the evaluation facility breaks the rules, then its termination by the CB will be done on the final day of the month provided that 3-month notice is given to the facility. The Singapore scheme gives the CB right to revoke a certificate if sponsor, developer, or facility breaches the specified terms and conditions. Moreover, the CB can terminate the on-going certification process without the issuance of the certificate by giving notice.

## Standardization

The scheme should have some standardized bodies to validate legal compliance and ensure uniformity of assessments conducted by different evaluation facilities. According to UK, Netherlands, US, and Singapore scheme, the standards as mentioned in Tables 3–5 should be followed and compliance should be made with them.

## Capacity building

It is another key factor that deals with arrangement of trainings to develop skillset for conducting evaluations according to CC. The US scheme has arranged a number of workshops to develop the technical and administrative skillset to conduct evaluations complying with CC. Similarly, Singapore hosted an international conference in October 2019.

## Strategic guidance

The guidance documents play a pivotal role in implementing the scheme properly. All four schemes provide some documents and forms, which can assist in developing and

implementing a scheme properly. The UK and Netherlands provide templates of technical reports, security manuals and meeting minutes. Along with these templates, the US also provides policy documents and NIST handbook to further ease the procedure. Similarly, UK presents a format for meeting minutes, technical reports, and security policy complying with ISO 27001 and ISO 27002.

### Structure

The structure and hierarchy of roles help in managing evaluation tasks. The US presents the structure of the scheme in a detailed form. Based on it, the scheme should has five levels of individuals in hierarchy, whereas the UK, Netherlands, and Singapore do not suggest such a structure.

## BEST PRACTICES TO BE FOLLOWED WHILE ESTABLISHING A NEW COUNTRY SCHEME

By the help of detailed analysis of four schemes presented in Tables 2–6 following best practices are proposed to conduct security evaluation of IT products in accordance with CC. Moreover, a backward comparison of the proposed best practices with the surveyed schemes is presented in Table 7 to demonstrate a better approach for establishing a new country scheme.

### With respect to the evaluation facility

**Hierarchy in Evaluation Facility's Role:** It is suggested to decompose the evaluation facility's role into multiple components so that role-based access control and separation of duty can be ensured.

**Owner of the Evaluation Facility:** As specified in Table 3, the evaluation facility can be government owned or some private company owned but in either case, it must comply with the national laws.

**Requisite Compliance with International Standards:** Those international standards should also be searched for which provide guidelines related to the chosen categories of products as are being considered in US, Singapore, Netherlands, and UK schemes. Afterward, compliance with those standards should be made as a prerequisite for certification in order to ensure better security.

**Assurance of Information Sanitization:** Principle of information sanitization should be followed as highlighted in all four schemes. It demands the need to sanitize the sensitive information from the product, before making it public or handing over it to the developer respectively as mentioned in UK Scheme. Moreover, the information security policy for implementing the non-disclosure of proprietary information is also recommended as is being followed in US scheme. The secure infrastructure of evaluation facility should also be ensured for information sanitization which is referred Table 3.

**Availability of Security Manual:** For providing assurance that appropriate security has been implemented inside the evaluation facility, presence of security manual is mandatory as specified in Singapore Scheme.

**Table 7 Comparison of the proposed best practices with the surveyed schemes.**

| Parameter | US | UK | Netherlands | Singapore | Proposed Particles |
|---|---|---|---|---|---|
| **With respect to Evaluation Facility** | | | | | |
| Hierarchal role | ✗ | ✓ | ✗ | ✗ | ✓ |
| Ownership | ✓ | ✓ | ✗ | ✓ | ✓ |
| Compliance with Standards | ✓ | ✓ | ✓ | ✓ | ✓ |
| Assurance of Information Sanitization | ✓ | ✓ | ✓ | ✓ | ✓ |
| Availability of security manual | ✓ | ✓ | ✓ | ✓ | ✓ |
| Qualification levels of evaluator | ✗ | ✓ | ✗ | ✓ | ✓ |
| Legal agreements | ✗ | ✓ | ✓ | ✓ | ✓ |
| Withdrawal by evaluation facility | ✓ | ✓ | ✗ | ✓ | ✓ |
| Records and procedures meetings | ✓ | ✓ | ✓ | ✓ | ✓ |
| Conflict of interest | ✓ | ✓ | ✓ | ✓ | ✓ |
| Assurance continuity management | ✓ | ✓ | ✓ | ✓ | ✓ |
| **With respect to CB/Validator** | | | | | |
| Role defined | ✓ | ✓ | ✓ | ✓ | ✓ |
| Charges levied | ✓ | ✓ | ✓ | ✓ | ✓ |
| Interaction of CB with evaluation facility | ✓ | ✓ | ✓ | ✓ | ✓ |
| Assurance continuity management | ✓ | ✓ | ✓ | ✓ | ✓ |
| Notification of changed status | ✗ | ✗ | ✗ | ✓ | ✓ |
| Surveillance | ✓ | ✓ | ✗ | ✓ | ✓ |
| Publication responsibility | ✗ | ✗ | ✗ | ✓ | ✓ |
| Guidance from industry | ✗ | ✗ | ✗ | ✗ | ✓ |
| Withdrawal by CB | ✓ | ✓ | ✓ | ✓ | ✓ |
| Assurance of consistency | ✗ | ✗ | ✗ | ✗ | ✓ |
| Dispute handling | ✓ | ✓ | ✓ | ✓ | ✓ |
| Validation procedure | ✓ | ✗ | ✗ | ✓ | ✓ |
| Record sharing means | ✓ | ✓ | ✓ | ✗ | ✓ |
| Termination agreement | ✗ | ✓ | ✓ | ✓ | ✓ |
| Usage of international certification marks | ✓ | ✗ | ✓ | ✓ | ✓ |
| Steps for evaluation | ✓ | ✗ | ✗ | ✓ | ✓ |
| Requisite compliance with standard | ✓ | ✗ | ✓ | ✓ | ✓ |
| **With respect to Certification Scheme** | | | | | |
| Management of scheme | ✓ | ✓ | ✓ | ✓ | ✓ |
| Structure of scheme | ✓ | ✗ | ✗ | ✗ | ✓ |
| Components of scheme | ✓ | ✓ | ✓ | ✓ | ✓ |
| Participants of scheme | ✓ | ✓ | ✓ | ✓ | ✓ |
| Role of sponsor | ✓ | ✓ | ✓ | ✓ | ✓ |
| Meetings related to national or international issues | ✗ | ✓ | ✗ | ✗ | ✓ |
| Management of new interpretations | ✓ | ✓ | ✓ | ✗ | ✓ |
| Certificate validation period | ✓ | ✗ | ✓ | ✓ | ✓ |
| Types of evaluated products | ✓ | ✓ | ✓ | ✓ | ✓ |
| Nationally or internationally approved PPs | ✓ | ✗ | ✗ | ✓ | ✓ |

| Table 7 (continued) | | | | | |
|---|---|---|---|---|---|
| Parameter | US | UK | Netherlands | Singapore | Proposed Particles |
| Surveillance by scheme | ✗ | ✗ | ✗ | ✗ | ✓ |
| Assurance continuity management | ✓ | ✓ | ✓ | ✓ | ✓ |
| **With respect to Accreditation Body** | | | | | |
| Role of accreditation team | ✓ | ✓ | ✓ | ✓ | ✓ |
| Role of accreditor | ✓ | ✓ | ✓ | ✓ | ✓ |
| Accreditation procedure | ✓ | ✓ | ✗ | ✓ | ✓ |
| Surveillance | ✓ | ✓ | ✗ | ✓ | ✓ |

**Qualification Levels of Evaluator:** The evaluation facility's policy must define the qualification levels of an evaluator as defined in UK scheme so that the product's evaluation can be assigned to the respective evaluator based on his qualification accordingly.

**Legal Agreements:** Legal agreements like contract between sponsor and CCTL, certification agreement with CB, contract regarding sharing of technical reports as mentioned in subsection of Singapore and UK schemes, subsection of Netherlands scheme, and subsection of US scheme respectively, should be signed in order to avoid any conflict in the future. In case, the facility terminates the agreement with the CB, it should make the required procedures and formalities clear.

**Steps for Evaluation:** The steps followed for evaluation like pre-evaluation, check-in meeting, monitoring, and certification phase should be distinctly mentioned as specified in US scheme so that CB can ensure that appropriate steps are being followed.

**Tasks of Evaluation Facility:** It would be significant to mention the tasks of an evaluation facility like preparation of technical reports, regulation of information flow etc. as elaborated in Table 3 so that no essential task is missed by it.

**Withdrawal by Evaluation Facility:** There should be some defined period of notice, which is 3 months in case of UK scheme as defined in Table 3, given by the evaluation facility before withdrawing some agreement with the CB or the developer so that they can take appropriate actions accordingly.

**Records and Procedures Meetings:** Records and procedures meetings should be held by the evaluation facility with the CB as mentioned in Netherlands scheme, with the sponsor as described in Singapore Scheme, and with the validator as mentioned in US Scheme, in order to ensure the correct implementation of all procedures and to inform sponsor about the evaluation progress of the product respectively.

**Conflict of Interest:** The facility should ensure that none of its members get a chance to assist the sponsor or developer as highlighted in UK scheme in the evaluation of product and it should assign a time period after the termination of its employee to not serve any developer or sponsor.

**Assurance Continuity Management:** In case the modifications are made to the certified TOE, two verdicts can be given by the evaluation facility, which are major and minor revisions as specified in US scheme. In case of minor revision, the facility will reevaluate it

by itself while in case of major revision, it will not only reevaluate it but it will also forward the ETR to the CB.

## With respect to CB/validator

The important practices that should be followed by CB to perform successful evaluation are listed below.

**Role of CB:** It should be responsible for managing the operations of scheme by enforcement of defined policies in accordance with CCRA as described in UK scheme, issues certificates in making evaluation assessments, and provide licenses to evaluation facilities as defined in scheme of Netherlands. Moreover, it also develops PPs, promotes integrity of CC certificates, etc. which has also been clearly defined in Table 4.

**Charges Levied by CB:** In order to ensure the correct implementation of regulations, the CB should levy charges on the evaluation facility. Moreover, it should also charge for the procedures like, certification, re-evaluation, assurance maintenance, accreditation, etc. as defined in Table 4.

**Interaction of CB with Evaluation Facility:** Healthy interaction should exist between CB and evaluation facility so that they both remain at the same level in following the procedures. The hiring of staff for evaluation facility by the CB, management of documents to inform the CB regarding the evaluation facility's progress, and the CB's task to resolve the queries related to CC interpretation, etc., as mentioned in UK and US respectively, are the tasks which are included in the interaction between CB and evaluation facility.

**Assurance Continuity Management:** In case the modifications are made to the certified TOE, two verdicts can be given by the evaluation facility, which are major and minor revisions as described in US scheme. In case of minor revision, the CB will not perform any further action. In case of major revision, the CB will prepare the VR and will add the revised version of the product to the PCL.

**Notification of Changed Status:** As elaborated in Singapore scheme, the CB should notify the community of any changes to the status of approved CC laboratories.

**Surveillance by CB:** The CB should have the authority to conduct an assessment of the evaluation facility at any time, may it be announced or unannounced, to know about its operating conditions and the methodology it follows. This has been explained in Table 4.

**Publication Responsibility:** The CB should publish publicly-releasable certification reports as described in Singapore scheme and periodically update a validated products list.

**Guidance from Industry:** The CB can seek guidance from industrial experts, if required.

**Withdrawal by CB:** If the evaluation facility does not comply with the defined policies and procedures then the CB should have the right to withdraw its agreement by giving it a prior notice of some specified period of time. The specified time in case of UK scheme as mentioned in Table 4 is 3 months.

**Assurance of Consistency:** The CB should ensure consistency of CCTL evaluations across the scheme.

**Dispute Handling:** In case some dispute occurs between the CB and the scheme or the evaluation facility, there should be some third party (the third party in case of Netherlands

scheme is national court as specified in Table 4) which should resolve the dispute by keeping in view the interest of all stakeholders. This is distinctively mentioned in US scheme too.

**Validation Procedure:** For the validation of the evaluation facility, the required formal interaction between the involved or concerned entities, procedures, documentation, etc. should be previously defined. This requirement is referred from Singapore and US scheme of Table 4.

**Record Sharing Means:** For the secure transportation of reports and other documentation, the CB should define the policy for sharing record among multiple entities. The email policy should be defined in the case of UK scheme and certified email should be availed in case of Netherlands scheme.

**Termination Agreement:** The CB should be able to terminate the agreement based on the occurrence of unavoidable factors and should highlight those factors in a legal agreement with the evaluation facility as highlighted in Table 4.

**Usage of International Certification Marks:** As described in subsection of Netherlands scheme, the CB should sign a joint agreement with the sponsor of the certified product and the certified evaluation facility to specify the necessary conditions under which they are authorized to use international certification marks.

**Steps for Evaluation:** The CB should design its own method, as specified in subsection of US scheme, to ease the process of product's evaluation which can then be used by the evaluation facility.

**Requisite Compliance with Standard:** If some national standard exists then CB should comply with it, provided that the requirement is given by the scheme.

## With respect to certification scheme

Following are the some important role and responsibilities of scheme to conduct security evaluations with respect to CC.

**Management of Scheme:** There should be some government-owned body which supervises the operation of the scheme. This requirement has been inferred from subsection of US and UK schemes.

**Structure of Scheme:** There should be a specified hierarchy in the scheme in order to ensure the role-based access control and separation of duty. Moreover, introduction of hierarchy will help all people know their extent of role and authority.

**Components of Scheme:** The scheme should comply with regulatory bodies, international or national standards, legal requirements, etc. to match the national and international legal requirements. This requirement has been deduced from Table 5.

**Participants of Scheme:** The scheme has the right to decide the entities which can become the participants of scheme. The sponsor, developer, vendor, procurement body, accreditor etc. can be the participants as specified in subsection of UK scheme.

**Role of Sponsor:** The role of the sponsor should be decided by the scheme. Normally, the sponsor is responsible for supplying all documentation and the supporting material which may be required in the evaluation of the product, pays all dues on the behalf of the developer as defined in subsection of Netherlands scheme, and inform the evaluation

facility if the certified product is modified. Moreover, there can be some additional role of sponsor too but it depends on the decision of scheme.

**Meetings Related to National or International Issues:** The scheme should specify some time interval, which is 3 to 4 months in case of UK scheme, after which meeting should be held among all the involved entities in relation to a scheme to resolve the controversial issues, may these be regarding some interpretation, some need of adding a new rule to the scheme, etc.

**Management of New Interpretations:** The scheme should define the proper channel which needs to be followed for adding new interpretation, if accepted, to the scheme. That proper channel can consist of application for request of interpretation followed by meeting with group of experts to decide whether new interpretation should be introduced or not as specified in the subsection of Netherlands scheme.

**Certificate Validation Period:** The certificate validity's duration should be mentioned by the scheme to prevent unauthorized usage of certificate. Moreover, this duration will guide the evaluation facility and sponsor to get the laboratory and product recertified respectively. The certificate validation period for a product is 5 years and for a site is 2 years as mentioned in Netherlands scheme.

**Types of Evaluated Products:** Instead of making efforts to make the evaluation facility certified for all category of products, rather it should go for only those types of products whose percentage of development within the country or parent company is significantly greater in number. This practice is being implemented in different schemes with respect to countries which can be seen in Table 5. This will not only save time but it will also help in increasing the revenue that will be obtained from evaluation. Afterward, the evaluation facility should struggle to be proficient in the specified categories of IT products.

**Nationally or Internationally Approved PPs:** The scheme should define whether it will certify the products whose ST will be made on the basis of national PPs, international PPs, or both. Acceptance of national PPs is recommended in US and Singapore schemes.

**Surveillance by Scheme:** The scheme should be authorized to conduct surveillance of the evaluation facility, may it be announced or unannounced, depending upon the decision of scheme.

**Assurance Continuity Management:** In case the modifications are made to the certified TOE, two verdicts can be given by the evaluation facility, which are major and minor revisions as described in US scheme. The verdict of major or minor revision will be made on the basis of the guidelines and procedures defined by the scheme.

## With respect to accreditation body

The roles and responsibilities that the accreditation body should perform are enlisted below.

**Role of Accreditation Team:** The roles of accreditation team should be distinctively defined. It can be composed of technical manager, coordination manager, quality manager, etc.

**Role of Accreditor:** The role of the accreditor is to approve and verify the compliance of the evaluation facility with the scheme and the specified national and international standards.

**Accreditation Procedure:** The accreditor should consult all specified standards and procedures prescribed in the scheme to accredit labs. The accreditation procedure should involve proficiency testing, review of the management system of laboratory, on-site visits, etc. as mentioned in US scheme, to get complete assurance that the laboratory is operating as specified.

**Surveillance by Accreditation Body:** The accreditation body should either specify the time interval after which it will conduct surveillance to accredit the evaluation facility or should visit the facility without making the visit announced in order to validate the procedures being followed in the facility for evaluation of products. It is recommended that the first surveillance should be conducted after period of 6 months as done in UK scheme. It is also suggested to conduct the successive surveillance after a period of 1 year and to conduct full assessment after a period of 3.5 years.

## CONCLUSION

This paper performs comprehensive comparative analysis of four CC schemes including UK, US, Netherlands, and Singapore in order to identify the best parameters for defining functions, roles, and responsibilities of scheme, CB, evaluation facility, and accreditation body in context of implementation approach as per CCRA instructions. After performing the comparison of schemes, majority of the factors contributed to declare US as the better scheme. Some of those are; firstly, US was found to have the greater number of evaluated products spread over wide categories of IT systems. Secondly, it was found to be the only scheme with a definitive scheme structure including five components. Additionally, it certifies products whose ST is based on national and internationally approved PPs in contrast to the UK and Netherlands scheme which do not specify this need. Based on the implementation methodology and parameters, lastly this paper has proposed best practices and guidelines to provide support for certificate consuming to become certificate authorizing nations.

## OPEN CHALLENGES

The future research can be focused on the derivation of strategy for security evaluation of IT devices under CC evaluation paradigm and automation of an entire process for CC evaluation. Moreover, mapping of the requirements provided by CC to other prevalent standards can be done to derive a harmonized approach. Moreover, if one country develops access control devices while the other develops firewalls then the requirements to establish evaluation facilities for both the countries will be different. Therefore, the analysis of the types of products being developed inside countries can be done to derive the mechanism which dictates the requirements of an evaluation facility based on the type of products being developed within respective countries.

### Funding

This research is supported by the Higher Education Commission (HEC), Pakistan through its initiative of National Center for Cyber Security for the affiliated lab National Cyber Security Auditing and Evaluation Lab (NCSAEL), Grant No: 2(1078)/HEC/M&E/2018/707. The funders had no role in study design, data collection and analysis, decision to publish, or preparation of the manuscript.

### Grant Disclosures

The following grant information was disclosed by the authors:
Higher Education Commission (HEC).
National Center for Cyber Security for the affiliated lab National Cyber Security Auditing and Evaluation Lab (NCSAEL): 2(1078)/HEC/M&E/2018/707.

### Competing Interests

Haider Abbas is an Academic Editor for PeerJ.

### Author Contributions

- Maheen Fatima conceived and designed the experiments, performed the experiments, prepared figures and/or tables, authored or reviewed drafts of the paper, and approved the final draft.
- Haider Abbas conceived and designed the experiments, analyzed the data, authored or reviewed drafts of the paper, and approved the final draft.
- Tahreem Yaqoob conceived and designed the experiments, performed the experiments, prepared figures and/or tables, authored or reviewed drafts of the paper, and approved the final draft.
- Narmeen Shafqat conceived and designed the experiments, analyzed the data, authored or reviewed drafts of the paper, and approved the final draft.
- Zarmeen Ahmad conceived and designed the experiments, performed the experiments, prepared figures and/or tables, authored or reviewed drafts of the paper, and approved the final draft.
- Raja Zeeshan conceived and designed the experiments, analyzed the data, authored or reviewed drafts of the paper, and approved the final draft.
- Zia Muhammad performed the experiments, analyzed the data, prepared figures and/or tables, and approved the final draft.
- Tauseef Rana analyzed the data, authored or reviewed drafts of the paper, and approved the final draft.
- Shynar Mussiraliyeva analyzed the data, authored or reviewed drafts of the paper, and approved the final draft.

## Data Availability

The acronyms used in the article are categorized into two lists and are available in the Supplemental Files.

## Supplemental Information

Supplemental information for this article can be found online at http://dx.doi.org/10.7717/peerj-cs.701#supplemental-information.

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
