# Peer review of "A survey on common criteria (CC) evaluating schemes for security assessment of IT products"

_PeerJ Computer Science, doi:10.7717/peerj-cs.701_

## Round 0.1 · original submission · Major Revisions

Dear Authors,

Although the comments to your paper would lead to a revision, I have the following doubts.

The paper is very long and is proposed as a literature review. However, the number of referenced documents is limited. What does that mean? 1) there are no papers/documents on the subject; 2) you considered only the most relevant ones under some criteria? Please clarify.

Moreover, the paper seems a user manual for security practitioners. It would be interesting to know if it is going to be submitted as a white paper to some security authority. Also, the paper could better serve as a manual for security evaluation, rather than a paper for PeeJ.

For the time being, the decision is to review the paper and its scopes profoundly.

And as a piece of advice, I suggest you submit the contents to some international body working on security standards, or that you make your own patent on the method.

Lastly, I suggest you submit to a security journal.

Of course, we will consider with pleasure a resubmission, but it seems a waste of skill. The decision is yours.

Thank you for considering PeerJ.

Regards

Reviewer 1 ·

Basic reporting

Quite a long article (35 pages) written in clear technical English. Short reference to existing literature: lines 91-107 and only two pages of bibliography. Abstract and Introduction describe broad and cross-disciplinary issues of interest for a large audience. The paper body is written for field professionals and practitioners with a lot of acronyms (two acronym lists could help reading: a list of organizations/standards, and a list of activities/processes/tasks) most of them defined (ECR, VR, TRRT ...) but some not defined even if require the same level of field knowledge (PCI, PP ...). Some acronyms are first introduced (SER, OR in Table 1 between lines 228 and 229) but defined later (SER, line 364; OR, line 371). Most tables are split in different pages: the "continuation" indication is present in tables 1-4 but not in table 5. Even if Common Criteria is a well established area of research, most of the bibliography sources are quite recent. The title refers to "Evaluating IT Products" but does not mention that Common Criteria is for IT Security Evaluation that could be an element taken into consideration when deciding of reading or not reading the paper. Something such as: "A Survey on Common Criteria Evaluating Schemes for IT Security Products Evaluation" could give a clearer idea of the subject.

Experimental design

An extensive investigation is performed at high level of detail. To improve politically correctness lines 863-864 could be rewritten: "Best practices to be followed while establishing a new country scheme" or similar. The survey methodology is consistent with the full coverage of the subject and sources are adequately cited. To improve legibility of reader with limited knowledge of the matter but interested in the field, Figure 3 should be introduced and explained before line 181.

Validity of the findings

Being a review of four best practices, novelty is limited but the paper could be of help for people interested in improving evaluating processes by comparing different schemes. Conclusions are limited to a few lines (1046-1054) and do not help in giving an idea on which of the cited schemes appears to be more friendly and/or more effective. An indirect valuation of the best practice is given from line 863 to line 1045. As for future work (Open challenges) the description is again (as in the Introduction) at a very high level (strategy for security evaluation of cyber networks) in conflict with the body of the paper dealing with the security evaluation of individual products. Also examples of the classes of products that have passed through the Common Criteria evaluation process could be useful (the interested reader can find the areas of application and the number of the verified products in the Common Criteria web pages): access control devices, boundary protection, detection devices, smart cards, mobility, networks, etc.

Additional comments

I appreciated the paper and I would help the authors in reaching a broader audience. The paper gives an idea of the complexity of producing good software to students.

Reviewer 2 ·

Basic reporting

no comment

Experimental design

no comment

Validity of the findings

no comment

Additional comments

This research summarizes functional requirements, assurance levels and methodologies for the evaluation of IT products through Common Criteria (CC), a well-recognized international standard to ensure security functionalities. A comprehensive comparative analysis of four well-known schemes has been provided. There is no question on the significance of the presented area of research as this research work proposes CC evaluation framework along with best practices and guidelines to assist certificate consuming members to gain signatory status and carry out CC evaluation.

The manuscript is well structured and explains the proposed framework in reasonable style. However, there are few suggestions (mentioned below) that might help to improve the manuscript quality.

1). Tables are referenced out of order, i.e. Table 4 is referenced (line 195) before Table 1 (line 228). Same is true for Table 3 and Table 2.

2). The paragraphs starting from lines 715, 719, 726, 733 and 738 require proper headings such as Validity Period, Cost, Withdrawal Procedure, Certification Marks and Termination of Agreement etc. respectively.

3). As the manuscript has lot of acronyms, so addition of an acronyms table can improve the readability.

4). The manuscript also has some typos and grammatical problems and needs to be revised.

Reviewer 3 ·

Basic reporting

.

Experimental design

.

Validity of the findings

.

Additional comments

Summary: The paper deals with a detailed examination of the Common Criteria (CC) evaluation schemes for evaluating IT products. First the paper gives a quick overview of CC, by listing its objective and structure. Then it identifies, in a list of countries which all adopt and deploy CC to some extent, four countries that do so more and better than others and for which data are available. Subsequently, each of these four countries is examined in detail, with respect to how much and how well (with respect to the CC specification) they implement CC. This exam is carried out independently for each country. Starting from such exams, a comparison is carried out between these four countries, with respect to a series of parameters which reflect the requirements of CC. At last, from such a comparison the authors identify a best practice list of suggestions about how to implement effectively and efficiently a CC scheme.

Clarity of language: The paper is understandable, though quite long and detailed, and is written in reasonable english. I think however, that more attention could be paid to the language correctness; see the attached paper with some suggested corrections (not applied to the whole paper, as it is quite long).

Originality: The paper is quite analitycal and examines in detail the deployment of CC in some significant countries. This part may be interesting for the specialist reader, but is not really original. The original part is the best practice guideline list for realizing a good CC scheme, or for improving an existing one.

Comment: The paper is very analytical, as said, and in this sense it can be useful for those readers
who wish to have an overview of CC, and to study some cases where CC is applied with some extensiveness and success. This part of the paper does not seem to be significantly original, anyway. The originality is all in the proposal of a series of guidelines for deploying a good CC scheme. However, such a series of guidelines is obtained by extracting examples from the previous analysis. It is not easy, if ever possible, to evaluate its quality. I would have appreciated that, at least, the proposed scheme had been compared to those of the four chosen countries, to identify those parts where they diverge from the proposed one, or where a feature of the proposed one is simply not realized. Such a backward comparison might be useful to appreciate better the novelty and possibly the effectivess of the best practice suggestions provided by the paper.

---

## Round 0.2 · Minor Revisions

Dear Authors,

Thank you for revising the paper so thoroughly. It looks improved now.
I ask you for another effort. The paper is still too long and dense. The idea of introducing tables is good and should be extended. In order for a reader to find profitable material, it is necessary that this paper takes the form of a sort of "manual" or guide.

Also, did you find any current literature, meaning of 2021?

The paper should be slimmed and shortened.

Otherwise, I suggest proposing the paper to ACM Surveys, which now publish more frequently per year.

Let us know if you can slim down the paper in 4 weeks.

Best regards

The Editor

---

## Round 0.3 · accepted · Accept

Dear Author,

Your paper has improved significantly. The paper now is clear and well structured. It states clearly that your paper is a guide for security developers and analysts.

Consider only to revise the tables' format: these are too tiny. Some can become an annex. Others can be enlarged (e.g. Table 7) and stay in the text.

Final Details (which can be addressed while in production):

Analysis of Country level: explain why you have selected these Countries. Country-level is also used: uniform.

Congratulations upon acceptance!

MG FUGINI